# ROUTESPLAIN: TOWARDS FAITHFUL AND INTERVENABLE ROUTING FOR SOFTWARE-RELATED TASKS

## ABSTRACT

LLMs now tackle a wide range of software-related tasks, yet we show that their performance varies markedly both across and within these tasks. Routing user queries to the appropriate LLMs can therefore help improve response quality while reducing cost. Prior work, however, has focused mainly on general-purpose LLM routing via black-box models. We introduce Routesplain, the first LLM router for software-related tasks, including multilingual code generation and repair, input/output prediction, and computer science QA. Unlike existing routing approaches, Routesplain first extracts human-interpretable concepts from each query (e.g., task, domain, reasoning complexity) and only routes based on these concepts, thereby providing intelligible, faithful rationales. We evaluate Routesplain on 16 state-of-the-art LLMs across eight software-related tasks; Routesplain outperforms individual models both in terms of accuracy and cost, and equals or surpasses all black-box baselines, with concept-level intervention highlighting avenues for further router improvements.

## 1 INTRODUCTION

Large Language Models (LLMs) are being adopted for an increasing variety of software-related use cases, such as code generation and repair, test-case generation, and computer science question-answering. As the landscape of available LLMs continues to expand with LLMs offering varying capabilities and pricing tiers, a critical question remains underexplored: do different models perform better or more efficiently on different types of software-related queries?

If such performance variations exist and can be efficiently predicted, they could enable optimization through query-based routing in two scenarios: (a) expert routing for queries requiring specialized capabilities; (b) cost-effective routing when multiple LLMs can adequately answer the query. Prior work has explored routing for general-purpose LLM use cases (Mohammadshahi et al., 2024; Sikeridis et al., 2025; Ong et al., 2025), but typically focused only on simple code completion scenarios (Hu et al., 2024; Zhuang et al., 2025) rather than exploring the rich variability across and within software-related tasks. In this work, we investigate routing efficiencies in accuracy and cost across the diverse landscape of software-related queries, from code generation and repair to execution prediction and computer science question-answering.

Moreover, unlike general text tasks, software queries naturally decompose into concepts: high-level, interpretable characteristics such as programming language, task type, and domain. Therefore, whereas existing general-purpose approaches either rely on black-box models (Ong et al., 2025) or try to implicitly (Zhuang et al., 2025) or explicitly (Song et al., 2025) capture abstract model abilities, software-related task routing lends itself well to concept-based routing.

We propose Routesplain, the first LLM router specifically designed for software-related tasks. Building on concept bottleneck models (Koh et al., 2020), Routesplain extracts concepts from queries and routes based solely on these concepts. This approach provides routing rationales that are *faithful* (Jacovi & Goldberg, 2020), as the concepts accurately reflect the model's reasoning process, and *intervenable* (Koh et al., 2020), since the model's reasoning process is easy to edit at the concept level. Furthermore, we demonstrate that these interventions are *predictable*, with concept-level changes leading to expected routing decisions.

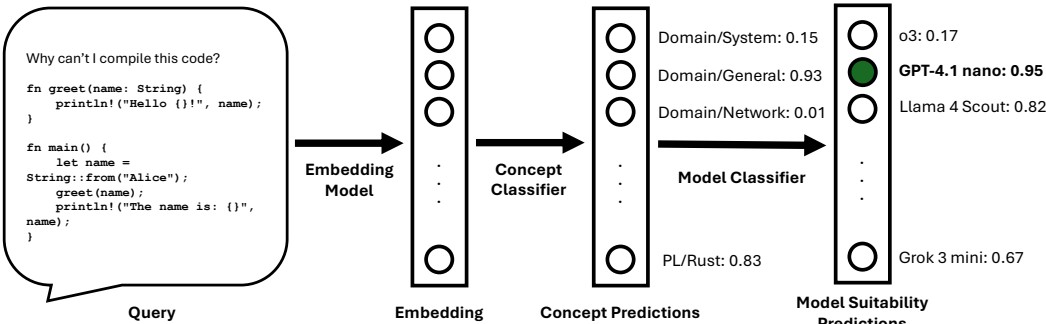

Figure 1: Overview of the Routesplain framework. Each query is embedded using a fixed embedding model. First, a concept classifier takes the contextualized embedding and projects it into the concept space, where each concept represents a high-level, interpretable characteristic of the query (e.g., task, domain, programming language). Then, a separate model classifier takes a concept-space input and outputs model suitability predictions. The query is routed to the most suitable model.

We evaluate Routesplain on a dataset comprising eight software-related tasks across 16 state-of-the-art LLMs including o3 and GPT-4.1. Routesplain outperforms individual models as well as existing black-box routing approaches KNN router (Hu et al., 2024) and EmbedLLM (Zhuang et al., 2025), achieving accuracy that matches a fully black-box MLP alternative across most cost regularization levels while providing strong interpretability and control through concept-level interventions. We also conduct ablation studies to determine the contribution of each concept group and counterfactual experiments to verify that changes in the concept space predictably impact the router's decisions. Lastly, we show that concept-level intervention can help pinpoint current routing performance limitations, specifically the difficulty and potential of accurately predicting query complexity.

Our contributions include: (1) the first comprehensive exploration of LLM routing potential within software-related use cases, demonstrating significant LLM performance variability across and within software-related tasks; (2) a novel LLM routing architecture that provides faithful and intervenable rationales for routing decisions without compromising performance compared to SoTA black-box approaches; (3) new methodologies for validating LLM routing correctness through concept ablation studies and counterfactual concept manipulation experiments.

## 2 RELATED WORK

Our work intersects two main research areas: LLM ensembling approaches for routing between models, and interpretability methods for understanding model decisions.

**LLM Ensembling** LLM ensembles are used in a number of ways to improve output quality and generation efficiency, including mixture-of-experts layers (Shazeer et al., 2017), speculative decoding (Leviathan et al., 2023), reranking LLM outputs (Ravaut et al., 2022), fusing LLM outputs (Jiang et al., 2023), cascading from weaker to stronger models (Varshney & Baral, 2022; Chen et al., 2024a), and routing. Routing approaches differ based on (1) learning paradigms, encompassing supervised (Mohammadshahi et al., 2024; Hu et al., 2024; Zhuang et al., 2025), unsupervised (Guha et al., 2024), or reinforcement learning (Lu et al., 2024; Sikeridis et al., 2025); (2) the size of the LLM pool, from a pair of strong and weak LLMs (Ding et al., 2024; Ong et al., 2025) to multi-LLM routing (Shnitzer et al., 2023; Šakota et al., 2024); (3) routing objectives, with single objective routing only focused on maximizing accuracy (Zhuang et al., 2025) or multi-objective routing focused on maximizing accuracy while minimizing cost (Mohammadshahi et al., 2024; Hu et al., 2024). To our knowledge, although previous work proposes routing to domain experts (Chai et al., 2024), Routesplain represents the first domain-specific LLM router. Particularly relevant to our work, recent approaches focus on implicit (Zhuang et al., 2025) or explicit (Song et al., 2025) learning of abstract model abilities. Instead, our routing strategy is based on high-level concepts/characteristics that make up the query. Chen et al. (2025b) propose tagging using a generative model and routing based on tag scores. While a compelling direction, a generative approach is both less efficient

Table 1: Model pricing and evaluation costs. Cost shows input/output token pricing. Average output length is measured in tokens per response. Total cost represents the expense of evaluating each model across our dataset collection. Models are grouped by cost-effectiveness tiers: high-cost (top), medium-cost (middle), and low-cost (bottom).

| Model | Cost ($/1M tokens) | Avg Output Length | Total Cost ($) |
|---|---|---|---|
| o1 | 15.00/60.00 | 147.79 | 557.59 |
| Grok 3 | 3.00/15.00 | 375.87 | 261.01 |
| Grok 4 | 3.00/15.00 | 376.66 | 261.47 |
| GPT-4o | 2.50/10.00 | 266.11 | 138.70 |
| Llama 3.1 405B | 4.00/4.00 | 220.99 | 91.40 |
| GPT-4.1 | 2.00/8.00 | 188.71 | 87.01 |
| o3 | 2.00/8.00 | 187.69 | 86.67 |
| o3 mini | 1.10/4.40 | 208.74 | 51.26 |
| o4 mini | 1.10/4.40 | 170.63 | 44.78 |
| GPT-4.1 mini | 0.40/1.60 | 199.58 | 18.07 |
| Grok 3 mini | 0.30/0.50 | 294.25 | 9.98 |
| Llama 4 Mav (fp8) | 0.15/0.60 | 242.39 | 7.77 |
| GPT-4o mini | 0.15/0.60 | 219.97 | 7.25 |
| Llama 3.3 70B | 0.13/0.39 | 221.77 | 5.20 |
| Llama 4 Scout | 0.08/0.30 | 306.20 | 4.70 |
| GPT-4.1 nano | 0.10/0.40 | 194.82 | 4.44 |

due to generation overhead than a lightweight classifier and less interpretable due to the large and unconstrained tag space, rendering routing decisions less faithful and harder to correct.

**Model Interpretability** Our work primarily focuses on ante-hoc explainability by adapting the well-known concept bottleneck model (Koh et al., 2020), widely used for image detection (Oikarinen et al., 2023; Panousis et al., 2024), to LLM routing. A popular alternative to ante-hoc methods are post-hoc methods such as LIME (Ribeiro et al., 2016), SHAP (Lundberg & Lee, 2017), model probes (Hewitt & Liang, 2019) and generating explanations using chain-of-thought (CoT) prompting (Wei et al., 2022) or training (Wang et al., 2024b; Yue et al., 2024). However, post-hoc methods lack the user's ability to intervene on concepts. Additionally, probes might capture representations that are not effectively used by the model (Belinkov, 2022). Similarly, CoT-produced rationales were found to be unfaithful (Turpin et al., 2023; Chen et al., 2024b; 2025a).

## 3 SOFTWARE-RELATED TASK ROUTING EVALUATION

In this section, we investigate the extent to which performance variations across LLMs on software tasks justify query-based LLM routing. We examine two scenarios: (1) expert routing, where certain LLMs significantly outperform others on specific queries, and (2) cost-effective routing, where multiple LLMs can answer queries but at different costs. We analyze both *intertask* variability (across different task types like code generation or repair) and *intratask* variability (within tasks based on structural features like programming language or domain), as well as how intratask patterns themselves vary across task types.

### 3.1 ROUTING DATASET CONSTRUCTION

Real-world software engineering requires expertise across diverse tasks beyond just code generation, including debugging, code understanding, test synthesis, and domain knowledge. We select tasks that represent these core software engineering capabilities with clear automated evaluation metrics, building upon and expanding the principle of holistic code evaluation (Jain et al., 2025) that examines multiple dimensions of programming competence.

We assemble popular datasets spanning these capabilities: BigCodeBench (Zhuo et al., 2025), a Python-based, open-domain, function-level code completion and natural-language-instruction-based code generation dataset; CRUXEval (Gu et al., 2024), a Python-based, function-level input and output prediction dataset; MMLU-ProX/Computer Science (Xuan et al., 2025), the computer science

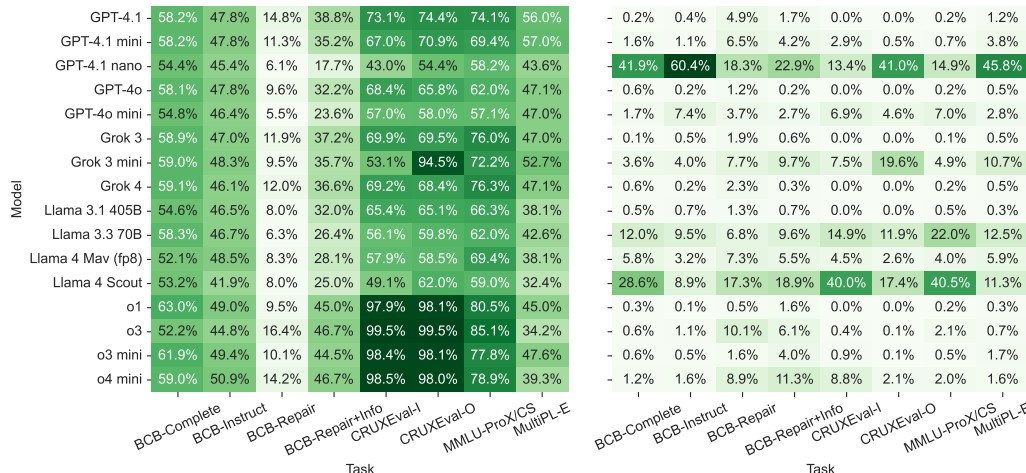

Figure 2: Intertask Performance. **Left:** 0-shot pass@1 accuracy of each LLM on each task. **Right:** Share of queries each LLM answered most cost-effectively per task. An LLM provides the optimal answer if it correctly answers a query at the lowest cost among all LLMs that answered correctly. For example, o3 provided the most cost-effective correct answer for 10.11% of BCB-Repair queries.

subset of a challenging multiple-choice question dataset translated to 29 natural languages; MultiPL-E (Cassano et al., 2023), a code completion dataset in 18 programming languages.

Recognizing the significance of code repair, similarly to prior work (Olausson et al., 2024; Zheng et al., 2025; Jain et al., 2025), we additionally construct repair tasks by reusing the incorrect responses of permissively licensed models on BigCodeBench. We filter the generated code snippets to include only non-trivially incorrect code, i.e., snippets that execute without crashing but fail to pass the test cases. Furthermore, we deduplicate the code snippets using basic string matching. We formulate two repair tasks: BigCodeBench-Repair, where we provide only the incorrect code snippet, and BigCodeBench-Repair+Info, where we additionally provide execution failure feedback (see Appendix E for prompt templates).

The entire dataset collection constitutes 38685 examples, spanning eight verifiable tasks: BCB-Complete (1140), BCB-Instruct (1140), BCB-Repair (5124), BCB-Repair+Info (5124), CRUXEval-I (800), CRUXEval-O (800), MMLU-ProX/CS (11890), and MultiPL-E (12 667). We evaluate 16 state-of-the-art LLMs that capture a wide range of capabilities and prices, across multiple families: GPT models (4.1 full/mini/nano, 4o full/mini), Grok models (3 full/mini, 4), o-series models (o1, o3 full/mini, o4 mini), and Llama models (3.1 405B, 3.3 70B, 4 Scout/Maverick fp8). We include a cost breakdown in Table 1. In line with previous literature (Zhuo et al., 2025; Rozière et al., 2024; Liu et al., 2023; Lai et al., 2023), we use greedy decoding to ensure reproducibility and measure 0-shot pass@1 accuracy.

## 3.2 INTERTASK PERFORMANCE

The left of Figure 2 shows that o-series reasoning models achieve the highest accuracy on most tasks, with o3 reaching 99.5% on CRUXEval tasks. The exception is multi-programming-language code completion, where GPT-4.1 mini achieves the highest accuracy at 57.0%. Nonetheless, the right of Figure 2 reveals that aggregate accuracy metrics obscure significant routing potential. Cost-effective models can handle many queries optimally: GPT-4.1 nano provides the most cost-effective solution for the majority of queries across BCB-Complete, BCB-Instruct, CRUXEval-O, and MultiPL-E tasks. Simultaneously, expensive models like o3 still provide optimal solutions for challenging queries (10.1% of BCB-Repair), demonstrating the value of expert routing for difficult tasks. Notably, even within the low-cost tier, different models excel on different tasks. While GPT-4.1 nano dominates most tasks, Llama 4 Scout provides optimal solutions for 40.0% and 40.5% of BCB-Repair+Info and CRUXEval-I queries respectively, compared to only 13.4% and 14.9% for

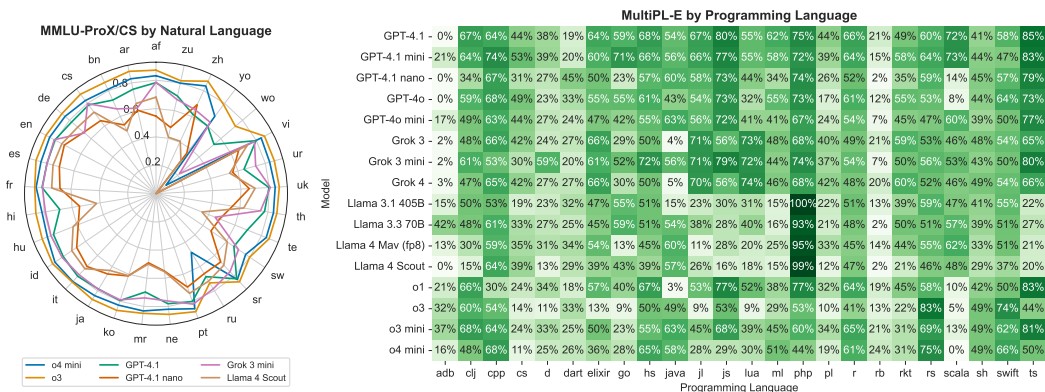

Figure 3: 0-shot pass@1 intratask accuracy. **Left:** Computer-science-related QA, stratified by the natural language of the query. **Right:** Multi-programming-language code completion, stratified by the programming language of the query. For radar plots, we select six representative models.

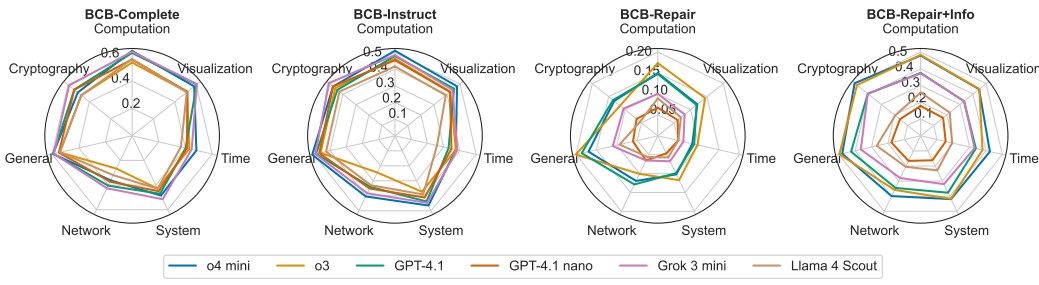

Figure 4: 0-shot pass@1 intratask accuracy comparison across open-domain code generation and repair tasks, stratified by the domain of the query, for six representative models.

GPT-4.1 nano on these tasks. This suggests that expert routing benefits exist not only between cost tiers, but also within them.

## 3.3 INTRATASK PERFORMANCE

We examine how LLMs perform on the same task under different conditions, focusing on natural language, programming language, and domain variations.

**Natural and Programming Language Variations** As shown on the left in Figure 3, LLM accuracy varies dramatically based on natural language. While o3 performs consistently well across most languages, it notably performs worse on Russian (similarly to the other OpenAI models GPT-4.1 and o4-mini) and is outperformed by Grok 3 mini. Lower-cost models match the accuracy of higher-cost models like o3 and GPT-4.1, especially on West-European languages like English, Spanish, French, or Italian. The performance gap between lower-cost and higher-cost models widens for South Asian languages like Hindi, Marathi, or Telugu. Notably, the accuracy of all but the pricier models o3 and GPT-4.1 drops significantly for Wolof, a low-resource language.[1]

In contrast to natural language patterns, programming language performance reveals more complex specialization patterns. As shown on the right in Figure 3, each model family demonstrates distinct advantages: Llama models excel at PHP (all achieving above 93% accuracy), GPT-4 models perform strongly in TypeScript, Grok models have advantages in Julia and Lua, and o-series models show strength in Rust and Swift. Notably, mini versions sometimes dramatically outperform their full-size counterparts–Grok 3 mini achieves 52 percentage points higher accuracy than Grok 3 on Java, and GPT-4o mini similarly outperforms GPT-4o on Scala.

---

[1]We adopt the geographic and resource classification of natural languages from Xuan et al. (2025).

**Domain-Specific Performance** Figure 4 reveals nuanced domain effects that vary across task types. Consistent with the intertask patterns in Figure 2, LLMs show much more similar performance to each other on original BigCodeBench tasks (code completion and instruction-based generation) compared to the derived repair tasks, where model performance becomes more stratified. Across all task types, models perform particularly well on generation- and computation-related domains but struggle more with network- and time-related tasks. However, the repair tasks exhibit notable differences depending on whether execution feedback is provided. Code repair with execution information reduces domain variability across all LLMs, while repair without execution feedback exacerbates these differences. Without execution feedback, models show stronger performance on general- and computation-related snippets but struggle with cryptography- and system-related code, despite not being disproportionately weak at generating such code initially. This pattern reveals interesting model-specific reversals: both o3 and Llama 4 Scout underperform on cryptography- and network-related repair tasks, where they are superseded by their cheaper alternatives o4 mini and GPT-4.1 nano, respectively.

These results demonstrate substantial intratask routing potential across multiple dimensions, with clear opportunities for both expert and cost-effective routing depending on domain, natural and programming language characteristics.

## 4 ROUTER FORMULATION

We propose a concept-based LLM router that bases routing decisions on interpretable, intervenable, high-level concepts. Before introducing our approach, we establish the general routing formulation.

The routing problem can be formulated as learning a function $f : \mathcal{X} \to \mathbb{R}^n$ that maps an input string $x \in \mathcal{X}$ to a vector $\boldsymbol{m} \in \mathbb{R}^n$, where $\boldsymbol{m}_i$ corresponds to the probability that LLM $i \in \{1, ..., n\}$ provides the most suitable response to $x$. The input $x$ is first projected into a $d$-dimensional contextualized embedding space by an embedding model, so the function to learn becomes $f : \mathbb{R}^d \to \mathbb{R}^n$, where $\boldsymbol{x} \in \mathbb{R}^d$.

A standard supervised approach trains a multi-label classifier that maps from input $\boldsymbol{x} \in \mathbb{R}^d$ to model suitability scores $\boldsymbol{m} \in \mathbb{R}^n$ given true model correctness labels $\boldsymbol{y} \in \mathbb{R}^n$ by minimizing binary cross-entropy loss with cost regularization (Equation 1). The cost term represents expected routing cost under the predicted probability distribution, where $cost(\boldsymbol{x}^i)$ is a normalized cost vector for input $\boldsymbol{x}^i$, and $\lambda$ controls cost sensitivity.

$$\hat{f} = \arg\min_f \sum_i (L_{BCE}(f(\boldsymbol{x}^i), \boldsymbol{y}) + \lambda f(\boldsymbol{c}^i) cost(\boldsymbol{x}^i)) \tag{1}$$

Our concept-based approach decomposes $f$ into two networks: $f(x) = g \circ h(\boldsymbol{x})$. We leverage existing dataset labels to create a $k$-dimensional concept space capturing natural language, programming language, domains, required libraries, and three complexity measures. The complexity measures represent the fraction of models that failed on each input, stratified by model type: reasoning complexity (fraction of reasoning models that failed), general complexity (fraction of non-reasoning models that failed), and total complexity (fraction of all models that failed). Reasoning and non-reasoning models form mutually exclusive categories within our 16-model evaluation set.

Network $h$ maps embeddings to concepts: $\mathbb{R}^d \to \mathbb{R}^k$, trained via Equation 2. Network $g$ maps concepts to model suitability $\mathbb{R}^k \to \mathbb{R}^n$, trained with cost-regularization via Equation 3. We train $g$ and $h$ independently rather than end-to-end. During inference, $\hat{g}$ receives predicted concepts from $\hat{h}$, enabling human intervention by editing concept predictions. We discuss the impact of concept-level correction in subsection 5.3.

$$\hat{h} = \arg\min_h \sum_i L_{BCE}(h(\boldsymbol{x}^i), \boldsymbol{c}^i) \tag{2}$$

$$\hat{g} = \arg\min_g \sum_i (L_{BCE}(g(\boldsymbol{c}^i), \boldsymbol{m}) + \lambda g(\boldsymbol{c}^i) cost(\boldsymbol{x}^i)) \tag{3}$$

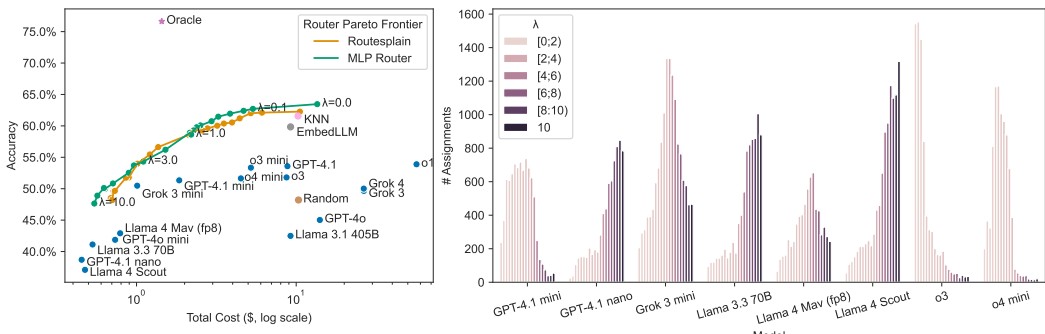

Figure 5: **Left:** Accuracy against cost of each individual model as well as all routers on the test set. For EmbedLLM, MLP router, and Routesplain, we show accuracy and cost averaged across 5 training runs. For Routesplain and MLP router, we display the Pareto frontier established by connecting the average router performance across multiple values of the cost regularizer $\lambda$. **Right:** The average number of times each model was assigned to an input by Routesplain, as we increase the $\lambda$ regularizer. We increase $\lambda$ by 0.1 on the interval $[0; 1)$ and then by 1 on the interval $[1; 10]$. We display only the top 8 most assigned models.

## 5 EVALUATION

We evaluate the efficacy of Routesplain using an 80/10/10 train/validation/test split on our dataset collection (see section 3). As the collection spans multiple programming and natural languages, we concatenate embeddings from two models: codesage-small-v2 (Zhang et al., 2024) and multilingual-e5-small (Wang et al., 2024a). To control for embedding quality, all baselines use the same embeddings. We retrain each model five times and report mean and standard deviation. Implementation and hyperparameter search details are provided in Appendix A. All experiments are conducted on a single Nvidia A100 80GB GPU.

All router models process the entire test set (3869 queries) in under 10 milliseconds, so latency and throughput are primarily determined by embedding time. Over 10 runs, embedding the test set with unoptimized, naive batching averages 36.96 seconds, yielding an average throughput of 104.68 queries per second, surpassing most proposed routing solutions by Ong et al. (2025).

### 5.1 HOW DOES ROUTESPLAIN COMPARE TO BLACK-BOX SOTA BASELINES?

We compare Routesplain against three black-box baselines: MLP router, KNN router (both standard baseline types proposed by Hu et al. (2024)), and EmbedLLM (Zhuang et al., 2025). For the MLP router, we use the black-box formulation in Equation 1. For the KNN router, we train separate binary classifiers for each model to predict correctness, routing to the model with highest predicted success probability. For EmbedLLM, we reimplement the algorithm as presented in Zhuang et al. (2025). All methods receive equal hyperparameter tuning effort. We also include random and oracle routers to establish theoretical bounds. To evaluate the performance of Routesplain and MLP router as cost sensitivity ($\lambda$) changes, we retrain each router with $\lambda$ incremented by 0.1 from 0 to 1, and by 1 from 1 to 10. This step size adjustment reflects the greater differences in cost-effectiveness among medium- and low-cost LLMs.

**Competitive Performance with Interpretability Benefits** The left of Figure 5 demonstrates that Routesplain achieves competitive performance while providing interpretability advantages. Routesplain Pareto dominates EmbedLLM, KNN router, random router, and all individual models across cost levels. Notably, Routesplain matches the black-box MLP router's performance at most regularization levels; the MLP router achieves statistically significantly higher mean accuracy only at $\lambda$ values of 0.0, 0.3, and 0.4, while Routesplain significantly outperforms the MLP router at $\lambda = 3.0$ (see the two-tailed $t$-test results in Appendix B). Moreover, Routesplain uses 15.65% fewer parameters (462K vs 547K) than the MLP router.

Table 2: Concept group ablation and intervention results across cost regularization levels. **Ablation:** Accuracy when removing each concept group during training (compared to baseline). **Intervention:** Accuracy when replacing predicted concept labels with gold labels at inference. Values show mean routing accuracy (%) with standard deviation over 5 runs.

| Concept Group | Concept Group Ablation Acc (%) | | | Concept Group Intervention Acc (%) | | |
|---|---|---|---|---|---|---|
| | $\lambda = 0.0$ | $\lambda = 0.1$ | $\lambda = 4.0$ | $\lambda = 0.0$ | $\lambda = 0.1$ | $\lambda = 4.0$ |
| *Baseline* | 62.27 (0.32) | 62.10 (0.55) | 53.98 (0.39) | 62.27 (0.32) | 62.10 (0.55) | 53.98 (0.39) |
| Tasks | 62.27 (0.32) | 62.02 (0.22) | 53.48 (0.41) | 62.30 (0.30) | 62.25 (0.63) | 53.89 (0.37) |
| Domains | 62.19 (0.32) | 62.37 (0.28) | 53.81 (0.32) | 62.27 (0.33) | 62.27 (0.68) | 53.92 (0.42) |
| Libraries | 61.69 (0.28) | 61.55 (0.32) | 52.56 (1.37) | 62.26 (0.32) | 62.30 (0.61) | 53.92 (0.39) |
| Natural Languages | 62.11 (0.16) | 62.15 (0.15) | 52.71 (0.60) | 62.26 (0.34) | 62.25 (0.65) | 53.90 (0.41) |
| Programming Languages | 60.43 (0.23) | 60.76 (0.42) | 51.08 (0.86) | 62.27 (0.33) | 62.13 (0.67) | 53.98 (0.41) |
| Complexity | 61.45 (0.29) | 61.28 (0.19) | 48.43 (0.69) | 66.32 (0.15) | 65.89 (0.17) | 57.02 (0.19) |

**Cost-Accuracy Tradeoffs** The left of Figure 5 illustrates the Pareto frontiers, showing how both Routesplain and MLP router navigate cost-accuracy tradeoffs. As cost regularization increases, both routers shift from expensive high-accuracy models to more economical alternatives while maintaining competitive performance. The right of Figure 5 shows how Routesplain's model assignments change with cost regularization. Higher $\lambda$ values shift assignments from expensive models (o3) toward mid-tier options (GPT-4.1 mini, o4 mini) and finally to the most cost-efficient models (GPT-4.1 nano, Llama 4 Scout). This pattern aligns with our dataset analysis findings in section 3, confirming that the router learns meaningful cost-accuracy relationships.

## 5.2 HOW DOES EACH CONCEPT CATEGORY CONTRIBUTE TO ROUTESPLAIN'S ACCURACY?

To understand which concepts contribute most to routing decisions, we conduct ablation studies by systematically removing each concept group and retraining Routesplain across three cost regularization levels ($\lambda = 0, 0.1, 4.0$). Results are shown on the left in Table 2.

**Concept importance varies significantly with cost sensitivity.** At high cost sensitivity ($\lambda = 4.0$), complexity concepts show the largest impact (-5.55 pp), aligning with the critical need to distinguish expensive versus cheap model requirements when cost matters most. Programming language distinctions matter substantially at all cost levels, reflecting the distinct performance profiles across models shown in subsection 3.3, but have greatest impact at the lowest cost level. Natural language distinctions show insignificant impact at low cost levels but meaningful effects (-1.27 pp) when cost is prioritized. This pattern reflects that expensive models handle most natural languages competently, while cheaper models struggle with low-resource languages (subsection 3.3). Library concepts are most important at high cost sensitivity and show greater significance than domain concepts, likely because they provide more granular, specific information than broader domain categories.

**Task categories show minimal ablation effects**, which we attribute to limited cross-dataset label variability in our current collection, as most tasks lack multiple degrees of freedom across natural languages, programming languages, and domains. This suggests promising directions for developing richer evaluation benchmarks with multi-language, multi-domain task variants.

## 5.3 WHAT DOES CONCEPT-LEVEL INTERVENTION REVEAL ABOUT ROUTING BOTTLENECKS?

Routesplain's concept bottleneck lets us perform a controlled counterfactual: at inference time we replace the router's predictions for a single concept group with ground-truth labels, leave every other concept prediction unchanged, and then rerun the router. Because only one factor is altered, any change in downstream accuracy can be attributed unambiguously to that group. The concept-level predictions quality by group is shown in Table 3, and the accuracy achieved after each oracle substitution across three cost regularization levels ($\lambda = 0, 0.1, 4.0$) is reported on the right in Table 2.

**Complexity concepts are the primary bottleneck for routing performance.** Providing oracle complexity labels raises routing accuracy by 3.04–4.05 pp across all cost sensitivities, with the largest gain at $\lambda = 0$. Substituting ground-truth labels for library or programming language concepts, however, does not yield any improvement, even though these groups are not predicted per-

Table 3: Concept prediction performance by group. We present accuracy, precision, recall, and F1 for binary labels and mean squared error and mean absolute error for complexity ratios, with mean and standard deviation over 5 runs.

| Concept Group | Acc (%) | Prec (%) | Rec (%) | F1 (%) | MSE | MAE |
|---|---|---|---|---|---|---|
| Tasks | 99.92 (0.02) | 99.79 (0.08) | 99.50 (0.07) | 99.64 (0.05) | — | — |
| Domains | 99.99 (0.01) | 99.66 (0.11) | 99.87 (0.05) | 99.76 (0.05) | — | — |
| Libraries | 99.99 (0.00) | 75.57 (0.56) | 75.30 (0.78) | 75.32 (0.67) | — | — |
| Natural Languages | 100.00 (0.00) | 99.98 (0.04) | 99.61 (0.12) | 99.79 (0.08) | — | — |
| Programming Languages | 99.90 (0.03) | 99.38 (0.50) | 91.79 (2.06) | 94.49 (1.77) | — | — |
| Complexity | — | — | — | — | 0.057 (0.005) | 0.176 (0.014) |

fectly (Table 3). Hence, errors in estimating query complexity propagate directly into routing mistakes, whereas errors in the other concept groups are largely absorbed by the second-stage classifier. Targeting complexity prediction (e.g., via a specialized sub-model) therefore offers the most leverage for improving routing accuracy, illustrating the diagnostic power of concept-level intervention.

## 5.4 DOES CONCEPT-LEVEL INTERVENTION LEAD TO PREDICTABLE ROUTING DECISIONS?

To test whether flipping a single concept produces the intuitive routing shift, we generate 1000 synthetic concept vectors for each of five source–target programming language pairs (e.g., PHP to Rust). All concept vectors describe the code completion task, English natural language, general domain, no libraries, and draw query complexity uniformly at random; the only difference between the paired vectors is that the programming language concept is set to the source language in one case and to the target language in the other. Passing both versions through Routesplain without cost regularization lets us measure how the selection of target-specific models responds to the language flip. Across the five pairs, changing the language concept increases the aggregated selection probability of the top three target-language models for the given language (as shown in Figure 3) by 37.03 percentage points and improves their average rank by 2.65 positions. The strong, consistent shift confirms that the router's decisions are directionally consistent in concept space: altering only the programming language concept predictably steers probability mass toward models specialized for that language, validating Routesplain's controllability through explicit concept manipulation.

## 6 LIMITATIONS

Routesplain has several practical constraints. The approach requires concept labels for training data and may lack representational power to capture nuanced query relationships not well-described by our concept taxonomy. Developing general-purpose concept spaces for diverse chat applications remains challenging, as our current concepts are tailored to code generation tasks. Additionally, current evaluation datasets lack diversity across multiple concept dimensions; for example, we leverage an execution-based open-domain Python dataset (BigCodeBench) and an execution-based simple standard library multi-programming language dataset (MultiPL-E), but no execution-based open-domain multi-programming language datasets currently exists. This limited cross-concept variability may constrain Routesplain's learning and transferability to out-of-distribution tasks. The system requires supervision and retraining when adding or removing models, adjusting concepts, or updating model pricing. However, retraining costs are minimal: Routesplain trains from scratch in approximately one minute on a single Nvidia A100, making iterative improvements feasible.

## 7 CONCLUSION

We introduced Routesplain, the first concept-based LLM router for software tasks, which matches black-box model performance while providing interpretable routing decisions. Our evaluation across eight tasks and 16 LLMs reveals significant routing potential and demonstrates that architectures enable diagnostic insights, such as identifying complexity prediction as the primary performance bottleneck, that are impossible with opaque systems. This work establishes interpretable routing as a viable approach for domain-specific LLM orchestration and opens directions for targeted improvements in routing system design.

## 8 REPRODUCIBILITY STATEMENT

To ensure reproducibility of our results, we have made several efforts to provide comprehensive implementation details and experimental specifications. Complete hyperparameter configurations for all router architectures (Routesplain, MLP router, KNN router, and EmbedLLM) are provided in Appendix A, including hidden dimensions, dropout probabilities, learning rates, and batch sizes determined through systematic grid search. Our evaluation uses publicly available datasets (Big-CodeBench, CRUXEval, MMLU-ProX/Computer Science, and MultiPL-E) with licensing information detailed in Appendix C. For our constructed repair tasks, we provide exact prompt templates in Appendix E and describe our filtering methodology for non-trivially incorrect code snippets in subsection 3.1. All experiments follow standard best practices with 80/10/10 train/validation/test splits, 5 independent training runs with reported means and standard deviations, greedy decoding for consistency, and 0-shot pass@1 accuracy evaluation. We specify the exact embedding models used (codesage-small-v2 and multilingual-e5-small), hardware requirements (single Nvidia A100 80GB GPU), and statistical testing procedures (two-tailed t-tests and Mann-Whitney U tests). The 38,685 evaluation examples span 16 state-of-the-art LLMs across eight software-related tasks, with model pricing and cost calculations transparently reported in Table 1. Our concept taxonomy and routing strategy are clearly defined in section 4, enabling replication of our concept-based routing approach.

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

## A  IMPLEMENTATION DETAILS

### A.1  ROUTESPLAIN

Routesplain consists of two separately trained classifiers. Each classifier is a two-layer MLP with GeLU (Hendrycks & Gimpel, 2023) and dropout. We conduct a separate hyperparameter grid search over plausible hidden dimensions, dropout probability, learning rate, and batch size for each classifier and select the combination that results in the lowest mean validation set loss across 3 random seeds. Namely, for the concept classifier, this is: hidden dimension = 256, dropout probability = 0.1, learning rate = 0.001, and batch size = 24. For the model classifier, this is hidden dimension = 176, dropout probability = 0.0, learning rate = 0.001, and batch size = 8.

### A.2  MLP ROUTER

The MLP router is a two-layer MLP classifier with GeLU and dropout. We conduct a thorough hyperparemeter grid search over plausible hidden dimensions, dropout probability, learning rate, and batch size and select the combination that results in the lowest mean validation set loss across 3 random seeds: hidden dimension = 384, dropout probability = 0.2, learning rate = 0.001, batch size = 8.

### A.3  KNN ROUTER

For the KNN router, we use the scikit-learn's KNeighborsClassifier (Pedregosa et al., 2011) and select the number of neighbors that results in the highest accuracy on the validation set: $k = 20$.

## A.4 EMBEDLLM ROUTER

For the EmbedLLM router, we reimplement the approach as described in Zhuang et al. (2025) and implemented in the accompanying GitHub repository. We conduct a thorough hyperparameter search over plausible hidden dimensions, model embedding dimensions, learning rate, and batch size and select the combination that results in the lowest validation set loss across 3 random seeds: hidden dimension = 512, model embedding dimension = 128, learning rate = 0.001, batch size = 32.

## B COMPARISON OF CLASSIFICATION ACCURACY BETWEEN ROUTESPLAIN AND MLP ROUTER

Table 4 summarizes the classification accuracy of Routesplain and the MLP router across varying levels of the cost regularizer $\lambda$. For each setting, we report the mean accuracy and standard deviation over five runs, and provide $p$-values from both a two-tailed $t$-test and a Mann-Whitney U test to assess statistical significance.

Our results demonstrate that the two models achieve comparable accuracy across almost all regularization levels. Statistically significant differences between their means are only observed at $\lambda = 0.0$, 0.3, 0.4, and 3.0 ($p < 0.05$ for both tests), with the MLP router performing slightly better at low $\lambda$ and Routesplain sometimes outperforming the MLP router at higher $\lambda$. These findings highlight the near parity in accuracy between Routesplain and the MLP router, and importantly, indicate that our interpretable approach can match or surpass the performance of a traditional black-box alternative, especially as cost regularization is increased.

| $\lambda$ | Routesplain Acc (%) | MLP Router Acc (%) | $t$-test $p$ | Mann-Whitney $p$ |
|-----------|---------------------|--------------------|--------------|------------------|
| 0.0 | 62.27 (0.32) | 63.46 (0.76) | 0.021 | 0.036 |
| 0.1 | 62.10 (0.55) | 62.71 (0.81) | 0.210 | 0.222 |
| 0.2 | 62.02 (0.38) | 62.40 (0.37) | 0.148 | 0.151 |
| 0.3 | 61.20 (0.50) | 61.94 (0.50) | 0.047 | 0.056 |
| 0.4 | 60.54 (0.27) | 61.46 (0.40) | 0.004 | 0.008 |
| 0.5 | 60.37 (0.39) | 60.74 (0.58) | 0.281 | 0.310 |
| 0.6 | 60.03 (0.38) | 60.07 (0.38) | 0.852 | 0.841 |
| 0.7 | 59.61 (0.36) | 59.78 (0.75) | 0.665 | 0.690 |
| 0.8 | 59.28 (0.55) | 59.28 (0.56) | 0.989 | 1.000 |
| 0.9 | 59.34 (0.49) | 58.90 (0.61) | 0.248 | 0.346 |
| 1.0 | 58.58 (0.59) | 58.61 (0.30) | 0.907 | 0.834 |
| 2.0 | 56.62 (0.43) | 56.20 (0.57) | 0.222 | 0.222 |
| 3.0 | 55.45 (0.60) | 54.32 (0.67) | 0.024 | 0.032 |
| 4.0 | 53.98 (0.39) | 53.70 (1.15) | 0.629 | 1.000 |
| 5.0 | 51.79 (0.64) | 52.53 (1.55) | 0.365 | 0.222 |
| 6.0 | 51.75 (0.61) | 50.83 (1.77) | 0.323 | 0.548 |
| 7.0 | 49.63 (1.10) | 49.98 (1.78) | 0.723 | 0.841 |
| 8.0 | 49.65 (0.38) | 50.11 (0.75) | 0.272 | 0.462 |
| 9.0 | 48.50 (0.88) | 48.90 (0.60) | 0.437 | 0.461 |
| 10.0 | 48.19 (1.06) | 47.64 (1.72) | 0.564 | 0.690 |

Table 4: Comparison of classification accuracy (%) between Routesplain and MLP router across varying values of the cost regularizer $\lambda$. For each $\lambda$, both models are retrained five times; we report the mean accuracy as well as standard deviation in parentheses. To assess statistical significance between model performances at each $\lambda$, we report the $p$-values from both a two-tailed $t$-test and the Mann-Whitney U test. This analysis provides insight into the differences in accuracy across different degrees of cost regularization.

## C LICENSING INFORMATION

We include the licensing information for the used datasets and embedding models in Table 5.

| Category | Name | License |
|---|---|---|
| Datasets | BigCodeBench | Apache 2.0 |
| | CRUXEval | MIT |
| | MMLU-ProX | MIT |
| | MultiPL-E | MIT |
| Embedding Models | intfloat/multilingual-e5-small | MIT |
| | codesage/codesage-small-v2 | Apache 2.0 |

Table 5: Licensing information for used datasets and embedding models.

## D  LLM USAGE

We used LLMs to help with grammar correction.

## E  BIGCODEBENCH-REPAIR PROMPT TEMPLATES

We include the prompt templates for BigCodeBench-Repair (Figure 6) and BigCodeBench-Repair+Info (Figure 7).

```
User:
I want to solve the following problem:
```
{prompt}
```

I produced the following code, but it does not work:
```python
{code}
```

Please provide the fixed code as a self-contained Python script
in a markdown code block.
```

Figure 6: Prompt Template for BigCodeBench-Repair.

```
User:
I was given the following problem to solve:
```
{prompt}
```

I produced the following code, but it does not work:
```python
{code}
```

Here are more details:
```
{additional execution failure information}
```

Please provide the fixed code as a self-contained Python script
in a markdown code block.
```

Figure 7: Prompt Template for BigCodeBench-Repair+Info.

