# OpenReview forum: "Routesplain: Towards Faithful and Intervenable Routing for Software-related Tasks"
_ICLR.cc/2026/Conference — ICLR 2026 Conference Withdrawn Submission_

### Official Review · Reviewer_H6to · 2025-10-28

**Soundness:** 4
**Presentation:** 4
**Contribution:** 4
**Rating:** 8
**Confidence:** 4

**Summary:**

This paper introduces Routesplain, the first LLM router for software-related tasks that can adaptively select a model that can balance cost and effectiveness. It leverages concept bottleneck models to achieve interpretable routing without sacrificing performance (compared with black-box counterparts).

**Strengths:**

1. Timely problem. The paper tackles a timely problem. Intelligently routing to use a cost-effective model can save a lot of money without sacrificing much performance.
2. Strong empirical evidence as motivation. In section 3, the paper performs an extensive study to show the inter-task and intra-task performance variance among models, highlighting the necessity of routing for software tasks.
3. Novel approach. The use of the concept bottleneck model (CBM) is novel and appropriate for the routing problem.
4. Solid evaluation. The authors compare Routesplain against baselines, including a black-box MLP router (that is not interpretable). Evaluation shows that Routesplain achieves better interpretability without losing performance.
5. Strong ablation and interesting findings. The paper performs ablation studies and delivered several interesting findings. For instance, Table 2 identifies the most salient features for the routing task (complexity, programming language). And the intervention study pinpoints the primary bottleneck for the framework is predicting query complexity.

**Weaknesses:**

1. Ambiguous or imprecise definition about “complexity”. The paper defines “complexity” as the fraction of models that failed on a given input. This definition is kind of circular (i.e., a query is complex because the strong models are required, and the strong models are required because it is complex).
2. Scalability of the concept space. The set of concepts is manually defined and tied directly to the labels available in the evaluation datasets. It remains unclear whether manual labeling and concept-based learning can scale to new tasks.

**Questions:**

1. Could you elaborate on your choice for the complexity definition? It’s reasonable to use the “complexity” metric, but more measurements can be used as well, e.g., existing code complexity metrics from software engineering perspective like cyclomatic complexity. Did you consider or experiment with complexity metrics like that? Do you believe the failure-rate-based label is learnable in a generalizable way?
2. What’s the cost of updating this framework, e.g., when adding a model to the pool, changing a model’s pricing, or updating the tasks? Although the retraining only takes 1 minute, what about the cost of data generation?
3. The intervention results in Table 2 show that fixing "Programming Languages" or "Libraries" yields no accuracy benefit, even though the concept classifier is not perfect for them (Table 3). You state the second-stage classifier "absorbs" these errors. This is an interesting finding. Does this imply that, for routing, a perfectly accurate concept classifier isn't actually necessary for non-complexity concepts? Does the model learn, for example, that "Rust" and "Go" are "similarly hard" and thus map them to a similar set of suitable models, making a misclassification between them less harmful?
4. The right of Figure 5 clearly shows the model assignment shifting as $\lambda$ increases. Does this $\lambda$ value need to be set manually, or do you envision a system where the user can provide a "budget" or "priority" (e.g., "fastest," "cheapest," "highest quality") that dynamically selects the appropriate $\lambda$ for the query?

---

> ### Author Response · Authors · 2025-11-17
>
> We sincerely thank the reviewer for their thoughtful and thorough engagement with our work. We greatly appreciate the recognition of our contributions and the insightful questions, which we address below.
>
> # Question 1: Complexity definition and alternative metrics
> We appreciate this thoughtful critique of our complexity definition.
>
> We acknowledge that defining complexity as failure rate is model-pool-dependent. However, we note that solve-rate-based definitions have precedent in the literature; for example, BigCodeBench uses model solve rate to help define its "Hard" subset (Zhuo et al., 2025). That said, we agree that more objective definitions like cyclomatic complexity would be valuable.
>
> The challenge is task coverage; while cyclomatic complexity, lines of code, or Halstead difficulty might be useful for narrow code generation tasks (like competitive coding), they cannot be easily applied to:
> 1. Code repair (bug difficulty may not correlate with code complexity)
> 2. Multilingual computer science QA (no code in the query)
> 3. Execution prediction tasks (very succinct code might contain complex library/API calls)
>
> We did experiment with incorporating standard solution-based metrics (cyclomatic complexity, lines of code, Halstead difficulty) for code generation tasks, but found they did not improve routing performance. We believe this is because tasks like BigCodeBench are complex due to their open-domain nature and the necessity to correctly use a variety of APIs instead of crafting long and complex coding constructs: this complexity is not well-reflected by the aforementioned metrics. We commit to including this analysis in the revision to demonstrate we explored objective complexity measures.
>
> One alternative we considered is using LLM judges to estimate task complexity, but this introduces challenges: (1) providing consistent instructions across diverse task types; (2) no guarantee of label quality; (3) additional inference cost. We believe our failure-based proxy, while imperfect, provides a learnable and generalizable signal across all task types in our evaluation.
>
> # Question 2: Cost of updating the framework
>
> The update costs vary by scenario:
> 1. Changing model pricing: Effectively free, since we simply update cost labels in the dataset and retrain (< 1 minute).
> 2. Adding a model to the pool: The primary cost is evaluating the new model on our dataset (38,685 examples). While non-trivial, this is a one-time cost and feasible even for expensive models in real-world deployments. For context, evaluating all 16 models in our pool cost ~$1,600 total (Table 1). Adding one model would be a fraction of this.
> 3. Updating tasks or concepts: Updating or adding new concepts would require regenerating concept labels for new data, which could involve manual annotation or automated extraction depending on the task. Updating or adding new tasks would involve reevaluating the model pool on newly added problems, incurring additional cost. As mentioned above, however, we believe that this one-time cost is feasible. We also envision the routing dataset remaining relatively compact, as users would curate a representative sample rather than continuously expanding it.
>
> In practice, the one-time evaluation cost is likely negligible compared to ongoing inference costs in production systems routing millions of queries.
>
> # Question 3: Why fixing PL/library predictions doesn't improve accuracy
> We agree with the reviewer's hypothesis and would be happy to include this discussion in the revision.
>
> We believe two distinct mechanisms explain why these concept errors are absorbed:
> 1. For programming languages: Misclassifications likely occur between syntactically similar languages where models perform similarly anyway. For example, Figure 3 (right) shows that models performing well on TypeScript also excel at JavaScript. Even if the concept classifier confuses these languages, the routing decision remains approximately optimal because the same models are suitable for both.
> 2. For libraries: Libraries add granularity to the domain concept but may lack much additional discriminative power. In BigCodeBench (Zhuo et al., 2025), (a) only popular domain-specific libraries have been chosen; (b) specific API usage is likely a larger challenge than models' general familiarity with the library itself. Libraries might therefore help refine routing decisions at the margin but aren't strong differentiators like complexity or programming language.
>
> This, as the reviewer suggests, is an interesting finding: for routing, a perfectly accurate concept classifier may not be necessary for all concepts, only for those with high discriminative power (complexity, in our case). We will add this discussion to the revision.
>
> (continued in the next comment due to length limitations)

---

> ### Author Response · Authors · 2025-11-17
>
> (continued from previous comment)
>
> # Question 4: Dynamic λ selection based on user priorities
> This is a great suggestion for practical deployment. Currently, each router instance is trained with a fixed λ value. However, because the router is so small (462K parameters) and inexpensive to train, we can train multiple router instances at different λ values (e.g., λ = 0.1 for "highest quality," λ = 4.0 for "balanced," λ = 10.0 for "cheapest").
>
> At inference time, the user provides a priority ("highest quality", "balanced", or "cheapest"/"fastest"), which maps to a specific λ value, and the system simply routes the query through the corresponding router instance. This requires no additional computation beyond selecting the appropriate model weights.
>
> We believe this is a promising direction for user-facing deployment and would be happy to discuss this possibility in the revision. Thank you for this suggestion!
>
> # On the weakness regarding complexity definition:
> We appreciate the reviewer raising the circularity concern. We acknowledge in Question 1 above that our definition is model-pool-dependent, and we have explored more objective alternatives. We will revise the paper to more explicitly discuss this limitation and include our experiments with alternative complexity metrics to show we have carefully considered this issue.
>
> # On the weakness regarding scalability of concept space:
> This is a fair concern. Our current concept set is tied to available dataset labels. For new tasks without such labels or for expanding the current concept space, the concept definition would require either:
> 1. Manual/LLM-assisted/LLM-based annotation
> 2. Automated extraction (if concepts can be inferred programmatically)
>
> We note that the one-time labeling effort by the router provider would likely happen regardless, as such labels might be leveraged for logging to monitor routing performance. However, we acknowledge this is an important direction for future work and would be happy to expand this discussion in the revision.
>
> We are grateful for the reviewer's careful evaluation and constructive feedback. We will incorporate the suggested analyses and discussions in our revision. Please let us know if any clarifications would be helpful.

---

### Official Review · Reviewer_GN5f · 2025-10-31

**Soundness:** 2
**Presentation:** 2
**Contribution:** 2
**Rating:** 2
**Confidence:** 4

**Summary:**

This paper proposes Routesplain, a routing system for large language models on software-related tasks. It selects the most suitable model by analyzing concepts extracted from queries, making the routing decisions both interpretable and intervenable.

**Strengths:**

- This paper successfully migrates the routing approach to the programming language scenario.
- The evaluation comprehensively covers 8 software tasks and 16 mainstream LLMs.
- It assesses performance variations of different models across these 8 tasks.

**Weaknesses:**

- The paper claims that the system is “interpretable” and “intervenable,” but these claims are largely conceptual. The so-called “interpretable” provided by the system appears to consist merely of a list of concept labels used in its decision-making. Furthermore, the described interventions do not meaningfully improve model performance and primarily serve to correct misclassifications made by the concept classifier. If the core purpose of the intervention is simply to rectify the system’s own errors, it remains unclear whether this should be considered a feature or a design flaw.
- The paper dedicates excessive discussion to LLM performance metrics, which comes at the expense of a clear exposition of its core methodological contributions. Given the rapid evolution of LLM performance, much of the reported data will quickly lose scientific relevance. The paper should shift its emphasis from these metrics to methodological contributions and technical novelty.
- Comparisons with black-box routing baselines remain superficial and lack in-depth mechanistic analysis. The paper does not clearly demonstrate in which specific aspects concept-based routing outperforms black-box methods in decision-making.
- Unlike directly querying a single fixed model, Routesplain introduces additional steps of concept classification and model selection, which incur significant latency for each query.  These steps add significant overhead for each query, which may be unacceptable in latency-sensitive real-world applications.
- The paper assumes users can simultaneously access and manage APIs for 16 top-tier LLMs, which is impractical in terms of both cost and control for most users. This severely limits the method’s generalizability and real-world applicability.
- The authors fail to clearly define the scenarios in which Routesplain is indispensable. For many common tasks, simply selecting a high-cost-performance general-purpose model (e.g., LLaMA 4 Scout) is already near-optimal. The marginal performance gains from introducing a routing system do not appear sufficient to justify the added complexity and cost.
- Key terms such as “interpretable” and “intervenable” are used without proper citations or with inappropriate references that do not strongly support the claims.
- The paper uses non-standard references such as “left of Figure 2” or “right of Figure 3,” which are inappropriate. Standard notation (e.g., Figure 2a, Figure 3b) should be used.

**Questions:**

- For code-related tasks, a straightforward approach to improving reliability is Pass@k (generating k outputs per problem). Please clarify why constructing such a complex routing system offers advantages over this direct and effective method.
- Given that many existing models already achieve strong performance across a broad spectrum of tasks, for common applications, selecting a top-tier model (e.g., LLaMA 4 Scout) is often near-optimal. The authors should clarify whether the marginal performance gains achieved through routing substantively justify the additional complexity and computational overhead introduced by the system.
- The authors should provide comparative experiments demonstrating whether Routesplain offers tangible benefits over simpler prompt engineering strategies (e.g., explicitly specifying programming language and task type in prompts).
- Since the paper focuses on the code domain, why were specialized code models such as Code Llama, DeepSeek-Coder, or StarCoder 2 not considered? Would including these models provide a fairer reflection of the state-of-the-art in this domain?

---

> ### Author Response · Authors · 2025-11-17
>
> We thank the reviewer for their engagement. We believe several key results may not have been sufficiently emphasized in our submission, and we address these concerns below to clarify our contributions.
> # On selecting a single top-tier model (e.g., Llama 4 Scout):
> This assertion contradicts our empirical findings in Section 3 and Figure 5 left.
>
> Llama 4 Scout provides the optimal answer for only 8.9% of BigCodeBench-Instruct queries and 11.3% of MultiPL-E queries. On our test set, Routesplain improves over Llama 4 Scout by 10.55pp (28% relative) at similar cost (+$0.07) and 26.37pp (71% relative) without cost constraints.
>
> Figure 5 left demonstrates that Routesplain Pareto dominates all individual models, improving over the next-best model (o3-mini) by up to 10.11pp. Section 3 exists precisely to establish that no single model is near-optimal across our diverse software tasks.
>
> # On intervention impact:
> Table 2 shows complexity intervention improves routing accuracy by 3.04-4.05pp, making Routesplain 3.18pp better than the MLP baseline (65.89% vs. 62.71% at λ=0.1).
>
> The reviewer states interventions "do not meaningfully improve performance," which appears inconsistent with these results. The reviewer then notes interventions "correct concept misclassifications" -- this is precisely the point. Intervention diagnostically reveals:
>
> 1. Complexity prediction errors account for 3-4pp loss in accuracy (actionable bottleneck)
> 2. Programming language and library errors are insignificant despite imperfect concept classifications
>
> Black-box routers cannot provide this diagnostic capability. Section 5.3 explicitly frames this as diagnostic ("What does concept-level intervention reveal...?"), not production enhancement. We consider this a core feature enabling interpretable debugging of routing decisions.
>
> # On 16-model pool being impractical:
> Our model pool is conservative relative to recent work and reflects real-world deployments:
> 1. Zhuang et al. (ICLR 2025, Spotlight) routes among 112 models
> 2. OpenRouter, Azure AI, AWS Bedrock, GCP Vertex provide unified APIs for 100+ models
>
> We did not arbitrarily select 16 models; these models are featured in enterprise deployments and are offered by the main AI coding products like Cursor and GitHub Copilot.
>
> # On latency concerns:
> Our routing adds negligible overhead. The router processes the test set (3,869 queries) in <10ms total. This is negligible compared to typical time-to-first-token (seconds) [1]. Including embedding, throughput is 104.68 queries/second, exceeding most SOTA routing solutions in Ong et al. (2025).
>
> # On specialized code models (CodeLlama, DeepSeek-Coder, StarCoder 2):
> These early 2024 models do not reflect current state-of-the-art. Our evaluation includes mid-2024 to 2025 models explicitly advertised for coding (GPT-4.1, o3/o4, Llama 3.3/4, Grok 3/4) and used in the main AI coding products like Cursor or GitHub Copilot. On a variety of benchmarks such as BigCodeBench (Zhuo et al., 2025) and EvalPlus [2], even our more efficient models such as GPT-4o mini or GPT-4.1 nano substantially outperform the suggested models.
>
> # On Pass@k:
> Pass@k is an evaluation metric, not a deployment strategy. It requires an oracle (e.g., test suite, human/LLM judge) to select from $k$ outputs -- this is often unavailable for real-world free-form queries. Even with an oracle, pass@k inflates cost and latency almost $k$-times vs. Routesplain's negligible cost and <10ms overhead.
>
> # On prompt engineering strategies:
> Prompt engineering and routing are complementary, not competing strategies. Advanced prompting (CoT, few-shot) improves individual models but does not eliminate cross-model performance variability -- as Gu et al. (2024) demonstrate, different models respond differently to prompting strategies like CoT. Routesplain is compatible with any prompting strategy and can be retrained in <1 minute if prompt changes alter model characteristics.
>
> Regarding adding programming language/task to prompts: we are unaware of evidence this benefits SOTA models that already condition on context. If the reviewer can point to specific work, we would be happy to include comparisons.
>
> # On comparisons with black-box baselines:
> We may not have clearly articulated what "black-box" means for mechanistic comparison. The baselines (KNN, EmbedLLM, MLP) cannot be decomposed to identify "specific aspects" of decision-making; this opacity is what defines them as black-box.
>
> We demonstrate: (1) matched performance (Figure 5 left); (2) diagnostic intervention capability (Section 5.3); (3) predictable concept-based control (Section 5.4). If the reviewer requests subgroup analysis (e.g., by programming language, domain), we can provide this. Otherwise, we would value specific guidance on what analysis would address this concern.
>
> (continued in the next comment due to length limitations)

---

> ### Author Response · Authors · 2025-11-17
>
> (continued from previous comment)
>
> # On Section 3 performance discussion:
> We may not have sufficiently motivated Section 3's role. As the first work on software-task routing, we must establish: (1) performance varies across models and tasks; (2) intra-task variation exists (e.g., programming language, domain); (3) routing opportunities exist for both cost-effectiveness and expert scenarios.
>
> Without Section 3, the natural question is "why not use one good model?" Our empirical analysis (~2 pages) provides the answer; our methodology and evaluation (Sections 4-5, ~3 pages) present the solution. We believe this balance is appropriate.
>
> Regarding future model improvements: performance heterogeneity will persist as the model landscape diversifies, making routing increasingly valuable. Our methodological contributions (concept bottleneck architecture, intervention analysis) remain relevant independent of specific model versions.
>
> # On interpretability beyond concept labels:
>
> We may not have adequately emphasized what concept-based routing enables beyond labels:
>
> 1. Faithful explanations: Routing uses only predicted concepts, not hidden correlations.
> 2. Diagnostic intervention (Section 5.3): Isolating that complexity prediction (not programming language or libraries) is the performance bottleneck.
> 3. Counterfactual validation (Section 5.4): Changing programming language concepts results in predictable changes in routing decisions.
>
> These capabilities distinguish our approach from black-box alternatives achieving similar accuracy. If the reviewer believes additional interpretability mechanisms would strengthen the work, we would value specific suggestions.
>
> # On "inappropriate citations":
> We would appreciate specific guidance, as we are uncertain how to address this concern.
>
> We cite Koh et al. (2020), the foundational ICML work on concept bottleneck models, which explicitly introduces and highlights concept-level interventions and how they strengthen interpretability of the concept bottleneck architecture (see paragraphs 4-5 of the introduction and Section 6). We also cite Jacovi & Goldberg (2020) for faithfulness. These are canonical references in the interpretability literature. Could the reviewer specify:
> 1. Which citations are inappropriate?
> 2. What alternative references would be suitable?
>
> # On figure references:
> We will adopt Figure 2a/2b notation in revision.
>
> # Additional References
> [1] Agrawal, Amey, et al. “Etalon: Holistic Performance Evaluation Framework for LLM Inference Systems.” arXiv:2407.07000, arXiv, 2 Sept. 2024.
>
> [2] Liu, Jiawei, et al. “Is Your Code Generated by ChatGPT Really Correct? Rigorous Evaluation of Large Language Models for Code Generation.” 2023, Thirty-seventh Conference on Neural Information Processing Systems.

---

### Official Review · Reviewer_KCF5 · 2025-10-31

**Soundness:** 2
**Presentation:** 2
**Contribution:** 2
**Rating:** 2
**Confidence:** 3

**Summary:**

This work propose a routing framework to send queries to experts for software-related task. The proposed method, RouteSplain, trains a concept router that map the input query to concepts via natural language, and then to a selected model in a pre-defined pool. The results showed that RouteSplain achieved a pareto front with a black-box MLP router, while offering more explainability.

**Strengths:**

- Routing queries to appropriate models to maximize performance and minimze cost is an important research problem.
- The proposed method achieved encouraging results, outperforming individual models.

**Weaknesses:**

- Suboptimal router formulation: a major drawback of RouteSplain is that while the router understands the queries, it does not understand the strengths and weaknesses of each component models. Thus, the trained router in RouteSplain is just a more advanced version of a count-based solution that counts how many times a model is selected for each combination of concept, and retrieve that during inference. This might explain why most routing strategies have similar result curves in Figure 5 Left. A more desirable solutions should also take into account each component experts during inference, before providing a routing score.

- RouteSplain has limited extensibility and out-of-distribution performance. Training and evaluation of RouteSplain follows the traditional in-domain setting, where a dataset is splitted into train-val-test sets, and the results are only reported in the test split. This strategy is quite limited in LLM evaluation. Can the trained RouteSplain generate to new benchmarks that share some similar software tasks such as CodeMMLU [A]. Despite being discussed in Section 6, I think that these aspects need to be investigated more thoroughly.

[A] Manh, Dung Nguyen, et al. "Codemmlu: A multi-task benchmark for assessing code understanding capabilities of codellms." ICLR (2025).

**Questions:**

See weaknesses.

---

> ### Author Response · Authors · 2025-11-17
>
> We thank the reviewer for their feedback. We believe we may not have explained some aspects of our method clearly enough, and we provide clarifications below.
>
> # On "Suboptimal router formulation - router doesn't understand model strengths/weaknesses":
> We believe this characterization misunderstands our architecture. Routesplain consists of two learned components (Section 4, Equations 2-3):
>
> 1. Concept classifier $h$: Maps queries to concepts
> 2. Model classifier $g$: Maps concepts to model suitability predictions
>
> The model classifier $g$ explicitly learns which models excel for which concept patterns. This is not "counting how many times a model is selected for each combination of concept". Instead, it is a neural network learning a continuous function from the concept space to model performance probabilities, trained on 30,948 examples with cost regularization (Equation 3).
>
> Section 5.4 empirically validates that $g$ has learned model-specific strengths: we generate concept vectors differing only in programming language (e.g., PHP to Rust) and measure routing changes. Changing the programming language concept increases selection probability of the top three target-language models by 37.03pp and improves their rank by 2.65 positions. This demonstrates the router learned, for example, that Llama models excel at PHP while GPT-4 models are stronger in TypeScript and o-series models outperform on Rust (Figure 3). These results directly contradict the claim that our router doesn't understand model strengths.
>
> If the reviewer envisions a different architecture that would "take into account each component experts during inference," we would appreciate specific technical guidance on what this means, as our model classifier already predicts each model's suitability given query concepts.
>
> # On "similar result curves in Figure 5 Left":
> We believe this observation supports rather than undermines our contribution.
>
> Our goal was to achieve interpretability and intervenability without sacrificing performance. Figure 5 left shows that Routesplain matches the black-box MLP router across most cost regularization levels while providing faithful explanations (by definition), diagnostic intervention capabilities (Section 5.3), and counterfactual validation (Section 5.4): capabilities the MLP router cannot provide.
>
> Routesplain Pareto dominates other SOTA black-box approaches (KNN router, EmbedLLM) in both accuracy and cost, demonstrating that our concept-based approach outperforms these methods while adding interpretability.
>
> Matching the strongest baseline (MLP) while providing interpretability is a success, not a limitation. If interpretability were "free" in terms of performance, there would be no tradeoff to study.
>
> # On "Limited extensibility and out-of-distribution performance":
> This concern applies to any supervised learning method, which is the standard paradigm for routing research.
>
> This concern applies to supervised learning methods, which represent the dominant approach in routing research. While unsupervised (Guha et al., 2024) and reinforcement learning approaches (Lu et al., 2024; Sikeridis et al., 2025) may offer different generalization properties, they introduce their own tradeoffs in terms of test-time efficiency, interpretability, and stability. All SOTA supervised routing approaches we compare against (KNN, EmbedLLM, MLP router) use task-specific training data and would face similar OOD challenges.
>
> However, we emphasize two mitigating factors:
>
> 1. Rapid retraining: As noted in Section 6, Routesplain trains in under 1 minute on a single GPU. Adapting to new benchmarks or task types therefore requires minimal computational overhead.
> 2. Concept transferability: Unlike black-box routers that learn opaque query-to-model mappings, Routesplain's concept-based approach may exhibit better transfer when new tasks share concept distributions (e.g., same programming languages, similar complexity profiles). This is an empirical question we acknowledge deserves investigation.
>
> Regarding evaluating on CodeMMLU specifically: we are unable to evaluate on this benchmark due to data licensing constraints beyond our control. Specifically, CodeMMLU is constructed from CommonCrawl and parsing copyrighted material from GeeksForGeeks and W3Schools; this does not comply with our copyright compliance policy.

---

### Note · Authors · 2026-01-03

I have read and agree with the venue's withdrawal policy on behalf of myself and my co-authors.